# Cellular and Humoral Response After Induction of Protection and After Finishing *Hymenoptera* Venom Immunotherapy

**DOI:** 10.3390/biom14121494

**Published:** 2024-11-24

**Authors:** Ajda Demšar Luzar, Matija Rijavec, Mitja Košnik, Urška Bidovec-Stojković, Jerneja Debeljak, Mihaela Zidarn, Peter Kopač, Peter Korošec

**Affiliations:** 1Laboratory for Clinical Immunology and Molecular Genetics, University Clinic of Respiratory and Allergic Diseases Golnik, 4204 Golnik, Slovenia; ajda.demsarluzar@klinika-golnik.si (A.D.L.); matija.rijavec@klinika-golnik.si (M.R.); mitja.kosnik@klinika-golnik.si (M.K.); urska.bidovec-stojkovic@klinika-golnik.si (U.B.-S.); jerneja.debeljak@klinika-golnik.si (J.D.); mihaela.zidarn@klinika-golnik.si (M.Z.);; 2Biotechnical Faculty, University of Ljubljana, 1000 Ljubljana, Slovenia; 3Faculty of Medicine, University of Ljubljana, 1000 Ljubljana, Slovenia; 4Faculty of Pharmacy, University of Ljubljana, 1000 Ljubljana, Slovenia; 5Faculty of Medicine, University of Maribor, 2000 Maribor, Slovenia

**Keywords:** *Hymenoptera* venom immunotherapy, sting challenge, biomarkers, serology, basophil activation test, follow-up

## Abstract

*Hymenoptera* venom allergy (HVA) is an IgE-mediated hypersensitivity reaction caused by *Hymenoptera* species stings (honeybee, vespid, or ant). The only effective treatment is *Hymenoptera* venom immunotherapy (VIT). Our study aimed to evaluate whether humoral and cellular biomarkers measured before, during, and after honeybee VIT are associated with the success of VIT, which was assessed by the response to a sting challenge one year after finishing VIT. In this prospective study, blood biomarkers of 25 patients undergoing honeybee VIT at the referral center in Slovenia were evaluated. A controlled honeybee sting challenge confirmed successful VIT in 20 of 25 (80%) patients. Honeybee venom (HBV) recombinant allergen profiles, evaluated before the treatment, were comparable between responders and non-responders. Longitudinal follow-up, up to 1 year after finishing VIT, showed that the immune responses do not differ significantly between patients with successful VIT and treatment failure. Those responses were characterized by decreased sIgE, tIgE, and BST, whereas sIgG4 levels increased. The basophil sensitivity also significantly decreases after VIT in both groups of patients. The analyzed biomarker which correlated considerably with treatment failure was higher basophil sensitivity to allergen stimulation before VIT. Similarly, systemic adverse events (SAEs) during the build-up phase of VIT correlated with treatment failure. Our study demonstrated similar sensitization profiles, and humoral and basophil immune responses to immunotherapy, in two different well-characterized groups of patients, one with successful VIT and the other with treatment failure. Notably, only high basophil sensitivity measured before VIT and SAEs during VIT were significantly associated with VIT failure, and both have the potential to be predictors of VIT failure.

## 1. Introduction

The *Hymenoptera* species are part of the biodiversity on which we all depend for our survival. They are present in almost every part of the world. Therefore, insect stings are relatively common, and individuals with hypersensitivity to *Hymenoptera* venom may experience allergic reactions, which vary greatly in severity from large local reactions at the sting site to systemic reactions. Insect venom anaphylaxis is the number one cause of anaphylaxis in the European population, accounting for 48.2% of all anaphylaxes, and the second cause in children (20.2%) [1]. The only effective treatment of patients with a high risk of systemic reaction due to *Hymenoptera* stings is venom immunotherapy (VIT), making 77–84% of honeybee allergic patients and 91–96% of vespid allergic patients tolerant to subsequent stings [2,3]. Thus, almost one in five patients remain at risk after finishing honeybee VIT.

VIT is a two-step procedure consisting of the build-up phase, in which venom concentration is increased in a stepwise fashion to the dose of 100 μg, and the maintenance phase, where 100 μg are administered at prolonged intervals [2]. The time to reach the maintenance dose varies from several hours to several weeks, depending on the protocol used, namely, the conventional (approx. 50 days), rush (five days), ultra-rush (couple of hours), or cluster protocol (approx. 30 days) [4,5]. In the case of a field sting during VIT or sting challenge (VIT treatment failure) or recurrent reactions to VIT injections during the maintenance phase, increasing the maintenance dose to 200 μg is recommended. As soon as the maintenance dose of immunotherapy is reached, short-term protection against *Hymenoptera* venom is established. The duration to achieve long-term tolerance varies among individuals. VIT is generally applied for 5 years [2,4,5,6,7,8]. However, in patients with known risk factors, such as a severe initial sting reaction or honeybee venom allergy with high exposure to further stings, clonal mast cell disease, and/or elevated basal serum tryptase (>11.4 μg/L), prolonged or even lifelong treatment is recommended [2,9,10,11]. To date, no biomarker of VIT efficacy has been established in clinical practice, making a controlled sting challenge a golden standard for evaluating venom protection or tolerance. However, the sting challenge is not widely used in clinical practice, as it could be potentially dangerous for patients; in addition to safety concerns, there are also logistic concerns [2].

Since *Hymenoptera* venom allergy (HVA) can present as a life-threatening anaphylactic reaction, identifying predictive biomarkers of VIT efficacy is a trending topic. Many studies have been focused on evaluating one or a few predictive biomarkers [12,13,14,15,16,17] Currently, the major laboratory parameter used to confirm sensitization to *Hymenoptera* venom is specific immunoglobulin E (sIgE) to venoms [2]. Yet there is no evidence that monitoring venom sIgE levels could predict the success of therapy.

Furthermore, specific immunoglobulin G4 (sIgG4) levels were studied. It can be used to follow immunological responses to immunotherapy. The observed increase in sIgG4 levels throughout allergen immunotherapy has been described decades ago [18], but their inhibitory activity declined after finishing VIT [15].

VIT can further modulate the humoral immune response by having an impact on mast cell function. Mast cell activity can be monitored by serum tryptase levels [19], and VIT was reported to decrease baseline serum tryptase (BST) concentration slowly but continuously over the time of the treatment, indicating reduced mast cell function or a decline in mast cell burden [20].

Basophils have a role in promoting humoral response and are also an important part of gaining venom tolerance acquired over the years of VIT. Previous studies have evaluated the activation of these effector cells using the basophil activation test (BAT) and concluded that VIT mainly decreases BAT sensitivity and not reactivity [21,22,23]. Further, a smaller study also showed that a decreased BAT sensitivity may correlate with VIT’s efficiency [21]. Basophil activation is also an independent risk factor for severe honeybee sting anaphylaxis and systemic adverse events (SAEs) during honeybee VIT [24].

Recent studies suggested the importance of component-resolved diagnostics in predicting VIT efficiency. The predominant sensitization to rApi m 10 (icarapin) is proposed to be associated with treatment failure as the icarapin seems to be poorly represented in some immunotherapy extracts [25,26,27,28].

Nonetheless, to this day, no biomarker has been established in clinical practice, making a controlled sting challenge the only method for evaluating venom tolerance indicating successful VIT [2].

Our study aimed to evaluate the immune response to honeybee VIT by measuring several humoral and basophil biomarkers correlating to VIT success. Having two very well-defined groups of patients treated with the same protocol in the same referral center, which resulted in either successful VIT or treatment failure according to the sting challenge, enabled us to evaluate the exact dynamics of below-mentioned biomarkers in both groups and investigate their differences. Thus, we analyzed in detail patients’ initial recombinant sensitization profiles and immune responses to VIT by characterizing venom-specific IgE, total IgE levels, BST, and venom-specific IgG4 follow-up. We also analyzed basophils’ response and basophils’ sensitivity to allergen stimulation and different clinical parameters, including systemic adverse events (SAEs) during VIT, in detail.

## 2. Materials and Methods

### 2.1. Study Population and Study Design

In this prospective study, we enrolled 25 patients with *Hymenoptera* venom allergy undergoing honeybee venom immunotherapy treated at the University Clinic of Respiratory and Allergic Diseases Golnik (Slovenia). Patients were diagnosed and selected for VIT according to the criteria narrated by the European Academy of Allergy and Clinical Immunology [2]. Patients underwent honeybee ultra-rush VIT as previously described [2,29,30,31], using HAL (Hal Allergy, Leiden, The Netherlands) or Stallergen (Stallergenes Greer, Baar, Switzerland) venom preparations. In the case of a systemic reaction during the build-up phase, a slower increase in the conventional dose was adopted. Treatment success was evaluated using a controlled sting challenge performed one year after finishing 5-year-long VIT [2]. Twenty patients were classified as successfully finishing VIT, and five patients as treatment failure. Treatment failure was characterized by a systemic reaction after a sting challenge (all had Ring and Messmer grades I or II reactions), and in all of those patients, the reaction was immediately treated with epinephrine. Whole blood samples were collected before the beginning of VIT, after the build-up phase (approximately day 5 of the treatment), and one year after finishing VIT before the controlled sting challenge. Samples were collected for all patients with successful VIT at all time points, whereas two samples in the time point after the build-up phase from the group of patients with VIT failure are lacking. Study group characteristics are described in detail in Appendix A. This study was approved by the Slovenian National Medical Ethics Committee (KME 0120-443/2020/3, approved on the 6th of November 2020). All patients gave their written informed consent to participate in the study.

### 2.2. Venom-Specific IgE and IgG4, Total IgE, and Basal Serum Tryptase

The serum concentrations of sIgE to honeybee venom [kIU/L] and total IgE (tIgE) [IU/mL] were measured on Immulite (Siemens Healthcare GmbH, Erlangen, Germany) according to the manufacturer’s instructions. Sensitization to honeybee venom was defined as an sIgE level of 0.35 kIU/L or higher. Furthermore, BST [ng/mL] and sIgG4 [mg_A_/L] to honeybee venom were measured on ImmunoCAP (Thermo Fisher Scientific, Waltham, MA, USA) according to the manufacturer’s instructions. Additionally, the sIgE/tIgE and sIgE/sIgG4 ratios were calculated.

### 2.3. Honeybee Venom Allergens

The determination of sIgE to honeybee venom allergen components was performed using the ImmunoCAP Phadia250 system (Thermo Fisher Scientific). The IgE concentrations against the following recombinant components were determined: rApi m 1 (Phospholipase A2), rApi m 2 (Hyaluronidase), rApi m 3 (Acid phosphatase), rApi m 5 (Dipeptidyl peptidase), and rApi m 10 (Icarapin). Sensitization was defined as IgE levels of 0.35 kIU/L or higher. The component-resolved diagnostics were performed for 24 patients. One patient was omitted from the analysis due to too low amounts of available serum.

### 2.4. Basophil Activation Test

The basophil activation test was performed as previously described in detail [21,24,32,33] using the FACSCanto II flow cytometer (BD Biosciences, Basel, Switzerland) according to the manufacturer’s instructions. Briefly, whole blood samples were incubated with a final honeybee venom concentration of 1 μg/mL, 0.1 μg/mL, 0.01 μg/mL, and 0.001 μg/mL (Hal Allergy, Leiden, The Netherlands) for 15 min at 37 °C. For the controls, the cells were exposed to stimulation buffer (with added IL-3) alone (negative control) or 0.55 μg/mL of anti-FcɛRI monoclonal antibody (mAb; Buhlman Laboratories, Basel, Switzerland) and 2 μM N-formyl-Met-Leu-Phe (fMLP; Sigma, St Louis, MO, USA) (positive control). Further, after stopping degranulation, anti-CD63 mAb, anti-CD123 mAb, and anti-HLA-DR mAbs (BD Biosciences) were added. Finally, whole blood probes were lysed, washed, fixed, and assessed for the surface expression of activation marker CD63 by flow cytometry. The threshold value of 15% of CD63-positive basophils was considered positive. Basophil sensitivity was determined as the venom concentration, giving a 50% maximum CD63% up-regulation (EC50). CDsens was calculated as the inverse value of the EC50 threshold multiplied by 100. The higher value for CDsens represents higher basophil sensitivity. Furthermore, the area under the curve (AUC) was calculated for every patient. Additionally, the percentage of change after VIT was calculated for CD63 basophil activation and AUC.

### 2.5. Data Analysis

The statistical evaluation of results was made in GraphPad Prism 10 software (version 10.2.3 for Windows; GraphPad Software, San Diego, CA, USA). The descriptive statistics are presented as medians and minimum–maximum range for measurement data and percentages for categorical data. Fisher Exact, Chi-square, Wilcoxon (for paired groups), or Mann–Whitney (for unpaired groups) tests were used as appropriate to determine statistically significant differences.

## 3. Results

### 3.1. Systemic Adverse Events During the Honeybee VIT Build-Up Phase Correlate Considerably with Treatment Failure

Even though VIT is generally well tolerated by patients, some individuals, especially those treated with honeybee venom [24], experience systemic adverse events, which mainly occur during the build-up phase of the treatment. Only two patients of 20 (10%) with successful honeybee VIT experienced SAEs during the build-up phase; both had mild skin reactions, appearing as itching of one or more body parts. On the other hand, all five patients (100%; *p*-value = 0.0055; Table 1) with VIT treatment failure experienced SAEs during the build-up phase of VIT; all had mild to moderate reactions with face erythema and/or itching. None of the patients experienced SAEs during the maintenance phase of VIT. Those observations suggest that SAEs during honeybee VIT represent an important indicator and risk factor for VIT treatment failure, which was also observed and described previously [2].

### 3.2. Recombinant Sensitization Profiles Are Not Associated with Honeybee VIT Immunotherapy Outcome

The progress of molecular allergology allowed us to identify the sensitization profiles of honeybee venom (HBV) allergens using component-resolved diagnostics [34]. The analysis of a panel of HBV allergens can improve diagnostic sensitivity compared with the use of rApi m 1 alone, can help in identifying additional major honeybee allergens, and might also be associated (in particular, Api m 10 sensitization) with honeybee treatment failure [25,28]. The individual HBV allergen sensitization profiles of patients with successful VIT and treatment failure are presented in Figure 1A and Figure 1B, respectively. Patients with VIT failure tend to be sensitized to more HBV allergens (three of five [60%] sensitized to ≥3 HBV allergens) than those with successful VIT (six of nineteen [31.6%] sensitized to ≥3 HBV allergens); however, the difference did not reach significance. Further, the proportions of sensitization to each recombinant venom allergen, namely rApi m 1, rApi m 2, rApi m 3, rApi m 5, and rApi m 10, showed slightly different sensitization profiles between the two groups of patients (Figure 1C). Patients with successful VIT were predominantly sensitive to r Api m 1 and rApi m 5, followed by rApi m 3, 10, and 2. On the other hand, patients with treatment failure were mainly sensitized to rApi m 1 and m rApi m 2, followed by rApi m 10 and rApi m 5. Again, there were no significant differences in the proportions of sensitizations to specific HBV allergens between those two groups. None of the patients with treatment failure were sensitized to rApi m 3.

### 3.3. Dynamics of Humoral Responses Are Not Reflective of Tolerance Induction

The relation between changes in antibody levels and clinical response to the allergen has been questionable for many years. Most HVA patients with antibody-mediated allergic reactions become tolerant after finishing VIT. A longitudinal evaluation of patients’ blood samples showed a transient increase after reaching the maintenance dose (25 patients before vs. 23 patients after reaching the maintenance dose; *p*-value = 0.0024) and a further significant decrease (25 patients before vs. 25 patients after VIT; *p*-value < 0.0001) in sIgE to honeybee venom levels throughout VIT (Figure 2A). Our results confirm the suggestion made by Varga et al., who concluded that the persistent decline in sIgE levels may play a role in obtaining long-term tolerance in VIT [15]. Similarly, Albanesi et al. suggested that sIgE levels in the follow-up were a possible predictor of the clinical efficacy of VIT [14]. Furthermore, we observed a constant and significant decrease in tIgE (*p*-value = 0.0147; Figure 2B).

A trend of higher median sIgE level before VIT seems suggestive of successful VIT compared to patients with treatment failure (6.54 kIU/L vs. 3.46 kIU/L), even though it does not show statistical significance. Similarly, it has been observed in responders to house dust mite immunotherapy with higher baseline levels of sIgE compared to non-responders [17].

According to sIgE and tIgE dynamics, we also observed sIgE/tIgE ratio changes after VIT in both groups of patients (Figure 2C). The sIgE/tIgE ratio was evaluated previously by Lorenzo et al. [16] and Wang et al. [17] regarding allergen immunotherapy. While the first study suggested a high specificity and sensitivity in distinguishing responders from non-responders to treatment, the second study found a lower performance in sIgE/tIgE ratio but nevertheless suggested a higher sIgE/tIgE ratio, in addition to other evaluated parameters, to be characteristic of responders. In our study, the sIgE/tIgE ratio was not proven to be a predicting factor of VIT efficiency.

The sIgG4 antibodies are known for their blocking abilities, and their role has been studied for many years. The blocking antibodies compete with IgE for allergens, thus inhibiting the IgE-mediated degranulation of the effector cells [29]. Recently, it was suggested that not the levels, but rather functional blocking activity, might better correlate to VIT effectiveness [12,13,14].

Interestingly, sIgG4 to honeybee venom shows no difference after the build-up phase of VIT, which is expected, as the time frame for their increase seems too short. Further, one year after finishing VIT, levels of sIgG4 were slightly and significantly higher than before starting VIT (*p*-value = 0.0301; Figure 2D). Accordingly, the sIgE/sIgG4 ratio decreases after the build-up phase of VIT on day five because of changes in sIgE and one year after finishing VIT because of changes in both IgE and IgG4 levels (Figure 2E).

The mast cell (MC) response can be monitored by serum tryptase as it reflects MC activity, and the BST may correspond to MC load or burden of MC disorders. It was previously shown that BST decreases over time in VIT [20]. In our longitudinal study, the BST follow-up shows a significant decrease after the build-up phase of VIT (*p*-value = 0.0007) and further after one year of finishing VIT (*p*-value < 0.0001; Figure 2F).

Importantly, neither venom sIgE and sIgG4, tIgE, and BST nor different ratios, including sIgE/tIgE and sIgE/sIgG4 during and after VIT, showed a difference between patients with VIT failure and successful VIT.

### 3.4. Basophil Sensitivity to Allergen Stimulation Differentiates Between Patients with Future Successful Venom Immunotherapy and Patients with Treatment Failure

Cellular response to allergen stimulation can be monitored by the basophil activation test. A significant decrease in basophil CD63 response was observed after finishing VIT in both HBV stimulation concentrations 1 μg/mL and 0.1 μg/mL (before vs. after; *p*-value < 0.0001; Figure 3A). These observations were described before in studies demonstrating that VIT decreases basophil response to allergens [21,22,23]. However, we could not confirm the result of a previous smaller study that showed that a higher decrease in BAT may correlate with VIT’s efficiency [21], as both patients with VIT failure and those with successful VIT showed similar BAT decreases. Additionally, the percentage of change in CD63 basophil response after VIT did not show significant differences between patients with successful and unsuccessful VIT (Figure 3B).

Further, CDsens was calculated from HBV stimulation concentrations of 1 μg/mL, 0.1 μg/mL, 0.01 μg/mL, and 0.001 μg/mL. A higher CDsens indicates higher basophil sensitivity. Those results showed that patients with treatment failure have markedly higher basophil sensitivity before the beginning of VIT than patients with successful VIT (median CDsens before VIT: treatment failure vs. successful VIT; 9733 vs. 1119; *p*-value = 0.0289; Figure 3C). That means the threshold at which allergen concentration basophils start to degranulate is lower for patients with treatment failure than those with successful VIT. Overall, of all assessed humoral and cellular markers, CDsens evaluated before starting VIT was the only biomarker differentiating between the two groups of patients.

Additionally, the AUC of basophil response to the same HBV concentrations, specifically 1 to 0.001 μg/mL, was also analyzed. Individual AUC calculations are shown in Appendix A. The median AUC decreases significantly one year after finishing VIT (Appendix A). However, no significant differences in AUC decreases were observed between VIT responders and non-responders. Correspondingly, no significant differences were observed between the two groups of patients in the percentage of change in the AUC after VIT (Appendix A).

## 4. Discussion

Evaluating clinical tolerance to *Hymenoptera* venom after venom immunotherapy is important for prognosis and identifying high-risk patients with possible treatment failure. Herein, we studied the immune responses of two well-characterized groups of patients whose clinical tolerance to venom was evaluated with a controlled sting challenge one year after finishing 5-year-long VIT. Our study aimed to assess immune biomarkers, such as sIgE, sIgG4, tIgE, BST, and basophil sensitivity, and characterize their dynamics during and after VIT in both groups of patients. Furthermore, the sensitization profiles of HBV allergens were evaluated before starting VIT. We found that immune responses characterized by a decrease in sIgE and tIgE levels, an increase in sIgG4 levels, and a decrease in basophil sensitivity are characteristic of VIT regardless of the VIT efficiency. The sensitization profiles were also not indicative of predicting VIT outcomes. Notably, only initially high basophil CDsens and adverse events during VIT correlated with VIT outcome and seem to be highly significant for patients with treatment failure.

The major clinical predictive factor of treatment failure in our study was SAEs during the first five days of VIT when the increase in dosing to the maintenance dose occurs. All patients with treatment failure experienced SAEs, like face erythema and/or itching. However, only a few patients with successful VIT experienced SAEs, which appeared as itching of one or more body parts.

The larger panel of relevant HBV allergens and the complexity of diagnosing HBV allergy with CRD resulted in the diverse sensitization profiles observed in our VIT-treated HBV-allergic patients, and similar observations were also found in numerous previous studies [25,26,27,28,34,35]. Further, some previous studies suggest that sensitization to some HBV allergens is associated with VIT treatment failure [25,28]. We could not confirm these observations as our study demonstrated similar sensitization profiles between responders and non-responders. Even though patients with VIT failure tend to be sensitized to more HBV allergens, the difference did not reach significance. However, additional studies, including larger studies focusing on outcomes data after finishing VIT, are needed to evaluate these possible associations further.

Herein, we showed that dynamics of humoral response do not reflect the induction of tolerance to *Hymenoptera* venom. Our longitudinal follow-up study showed a transient increase in sIgE levels during the build-up phase of the treatment and later, one year after finishing VIT, a significant decrease in both groups of patients. Our results confirm observations from previous studies of the decreased sIgE levels after VIT, with the main difference being that our study also differentiates between tolerance induction and treatment failure. Previously, Varga et al. suggested that the persistent decline in sIgE levels rather than IgG4 inhibitory activity may play a role in obtaining long-term tolerance in VIT [15]. Similarly, Albanesi et al. suggested sIgE levels in the follow-up as the best predictor of the clinical efficacy of VIT [12]. However, the threshold of IgE decline for efficient VIT is currently unknown, and we could not observe such a threshold in our study. Interestingly, in food allergy, it has been pointed out that the presence of sIgE and the titers of sIgE do not predict the severity of the reaction but rather the reaction probability [18]. Similar to VIT, prolonged grass pollen immunotherapy sIgE levels also reportedly decreased [20].

We also observed an overall decrease in total IgE levels. In concordance with the sIgE and tIgE dynamic, the sIgE/tIgE ratio also changes significantly after VIT. The decrease in sIgE/tIgE ratio has previously been suggested to be characteristic of responders to allergen immunotherapy [16,17]. In our study, the sIgE/tIgE ratio did not predict VIT efficiency. Interestingly, higher median levels of sIgE and tIgE before VIT seem to be a feature of patients with successful treatment. Further, more extensive studies are needed to confirm these observations.

Oppositely to sIgE and tIgE, the blocking IgG4 levels decreased transiently during the build-up phase of VIT and then stayed significantly increased one year after finishing VIT. An increase in sIgG4 may reflect changes in cytokine production by allergen-specific T cells. Accordingly, the sIgE/sIgG4 ratio changes significantly after VIT, indicating a shift from a Th2 to Treg immune response. The role of IgG4-blocking antibodies has been studied for many years. Recently, it was suggested that instead of the sIgG4 levels, the functional activity correlates to VIT effectiveness [12,13,14]. In allergen immunotherapy for ryegrass pollen, sIgG4 levels increase after each consecutive course of treatment, and this increase is associated with clinical improvement [36]. Similarly, a significant increase in IgG4 levels and a subsequent decrease in the sIgE/sIgG4 ratio was observed after finishing AIT for grass pollen, tree pollen, and cat allergen [37].

The dynamics of basal serum tryptase showed a slight but constant and significant decrease in median levels during and after VIT. The decline in BST was previously suggested in the longitudinal study of VIT made by Dugas-Breit et al. [20]. However, similar to other humoral biomarkers, it does not reflect tolerance induction; thus, it does not differentiate between patients with successful VIT and patients with treatment failure. Those observations may suggest the effect of VIT on mast cell function; however, additional studies are necessary to evaluate these speculations.

The cellular response to immunotherapy can be efficiently monitored by the basophil activation test. Basophil sensitivity significantly decreased throughout honeybee VIT, indicating that basophils are less prone to degranulation caused by HBV allergens. The cellular response was further studied by evaluating basophil sensitivity with CDsens. A higher CDsens indicates higher basophil sensitivity to HBV stimulation. CDsens decreased significantly throughout VIT. Importantly, measuring this before starting VIT seems to differentiate the two patient groups. Patients with treatment failure have significantly higher CDsens compared to patients with successful VIT. Different studies have previously suggested BAT for close monitoring of VIT [21,22,23,38]. Our results indicate that it could be used to predict which patients are at high risk for honeybee VIT failure before starting with VIT. Additional BAT studies are necessary to evaluate the possible routine clinical relevance of these associations.

Our study offers long-term insight into the immune dynamics and follow-up of two different groups of patients: patients with successful VIT and patients with treatment failure. The initial sensitization profiles were demonstrated to be relatively similar between responders and non-responders and are not indicative of predicting VIT outcome. Furthermore, we observed that the tolerance induction after finishing VIT does not reflect changes in humoral biomarkers. Nevertheless, we must bear in mind the possible decrease in the acquired venom tolerance each year after VIT termination [39] and that further studies evaluating patients more than one year after stopping VIT are needed. To note, since the sting challenge is highly unpleasant and, in the case of treatment failure, also life-threatening for the patient, a possible limitation of the study is the relatively small number of patients with treatment failure. Larger future studies should address this limitation and are needed to confirm our research findings.

Overall, we showed that VIT is characterized by a decrease in specific IgE, total IgE levels, BST levels, and an increase in sIgG4 levels, both in patients with successful VIT and treatment failure. In a study conducted in 1989 by Muller et al., who evaluated the predictive value of serological biomarkers after one year of VIT, a conclusion was made that no single antibody estimation or combination of various antibodies was predictive of the outcome of a sting challenge [40]. Even though many studies have tried to predict VIT efficiency, we suggest the same as Muller et al., which is that serological biomarkers do not reflect tolerance induction in VIT. However, SAEs during the build-up phase of VIT and the higher initial basophil sensitivity to allergen stimulation seem to correlate with the success rate of venom immunotherapy. Collectively, these findings expand our understanding of which mechanisms seem important or ambivalent for efficient VIT and which direction further VIT studies should follow.

## 5. Conclusions

The evaluation of *Hymenoptera* venom immunotherapy success depends on the controlled sting challenge which is highly unpleasant and, in the case of treatment failure, also life-threatening for the patient. Our study aimed to evaluate the dynamics of humoral and cellular markers in relation to VIT success.

Overall, we showed that VIT is characterized by a decrease in specific IgE, total IgE levels, BST levels, and an increase in sIgG4 levels, both in patients with successful VIT and treatment failure, suggesting dynamics of humoral responses are not reflective of tolerance induction. Further, initial recombinant sensitization profiles are not associated with honeybee VIT immunotherapy outcome. Notably, only high basophil sensitivity measured before VIT and SAEs during VIT were significantly associated with VIT failure, and both have the potential to be predictors of VIT failure.

## Figures and Tables

**Figure 1 biomolecules-14-01494-f001:**
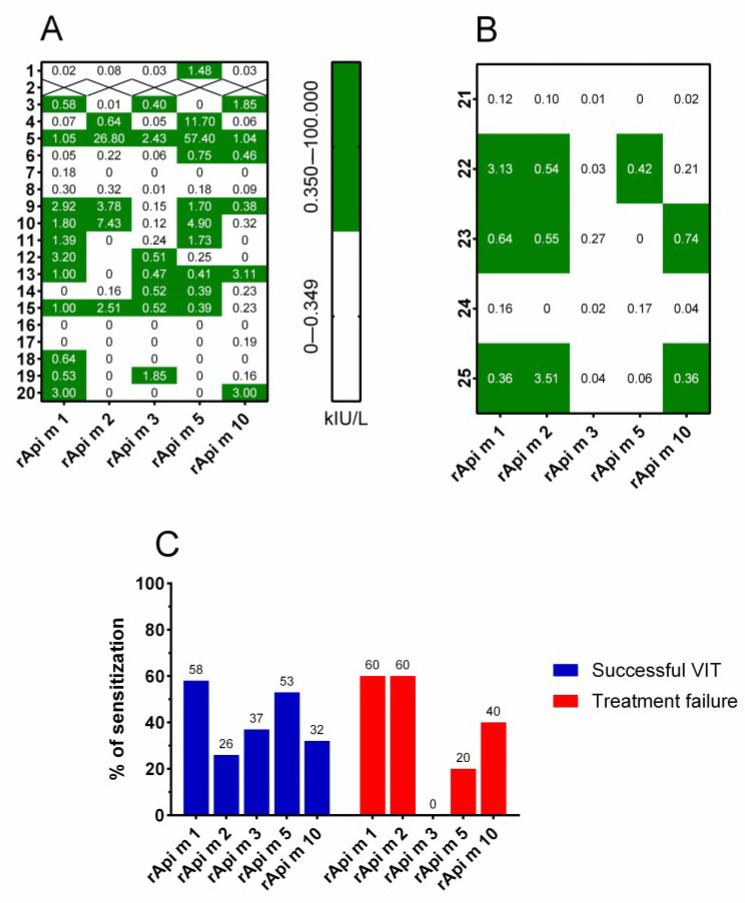
Sensitization profiles of HBV allergens evaluated before the beginning of venom immunotherapy. Individual sensitization profiles of (**A**) patients with successful VIT and (**B**) treatment failure. (**C**) Proportions of patients sensitized to each of the evaluated components of honeybee venom in two different groups of patients treated with honeybee venom immunotherapy: successful (blue) and treatment failure (red). Patient number 2 was omitted from the analysis due to too low amounts of available serum. HBV, honeybee venom; VIT, venom immunotherapy.

**Figure 2 biomolecules-14-01494-f002:**
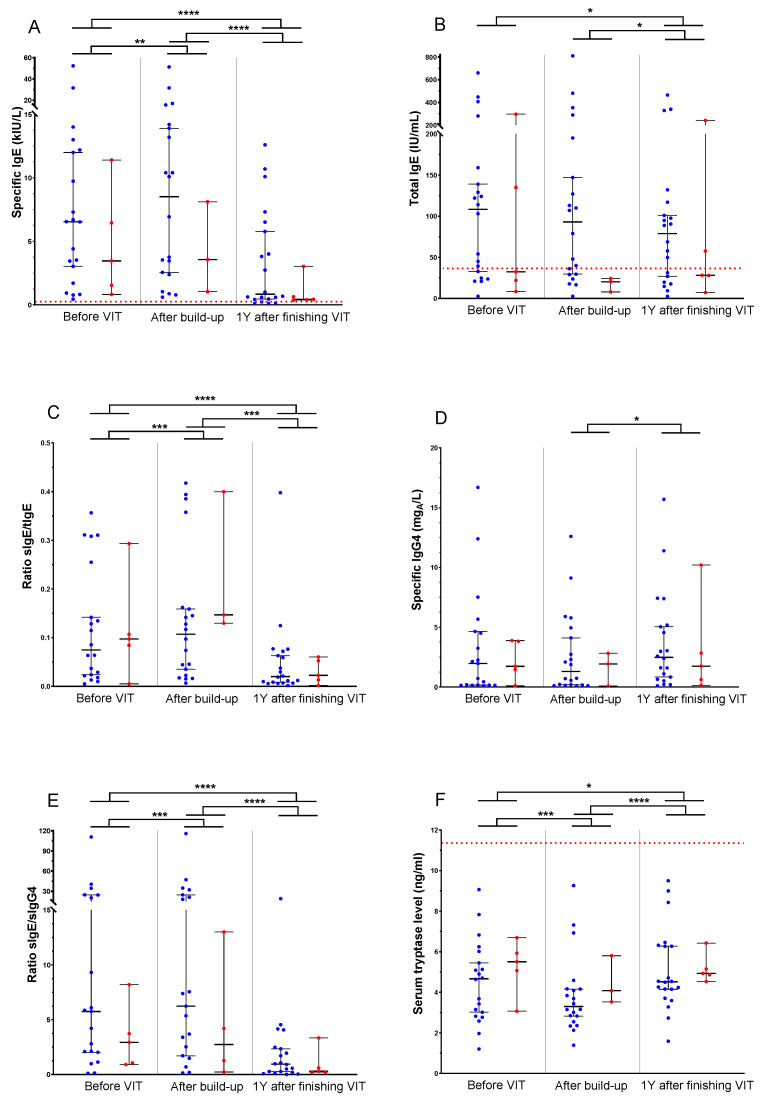
Timeline of humoral response in two different groups of patients treated with honeybee venom immunotherapy: successful (blue) and treatment failure (red). Those parameters were evaluated before the beginning of VIT, after the build-up phase of VIT (approximately on day five), and one year after finishing VIT. The dotted red line indicates reference values of 0.35 kIU/L and 100 IU/mL for sIgE and tIgE, respectively. Levels (median, range) of (**A**) specific honeybee IgE, (**B**) total IgE, (**C**) sIgE/tIgE ratio, (**D**) specific honeybee IgG4, (**E**) sIgE/sIgG4 ratio, and (**F**) serum tryptase at different time points during the treatment. VIT, venom immunotherapy; *, statistical significance *p*-value ≤ 0.05; **, statistical significance *p*-value ≤ 0.01; ***, statistical significance *p*-value ≤ 0.001; ****, statistical significance *p*-value ≤ 0.0001.

**Figure 3 biomolecules-14-01494-f003:**
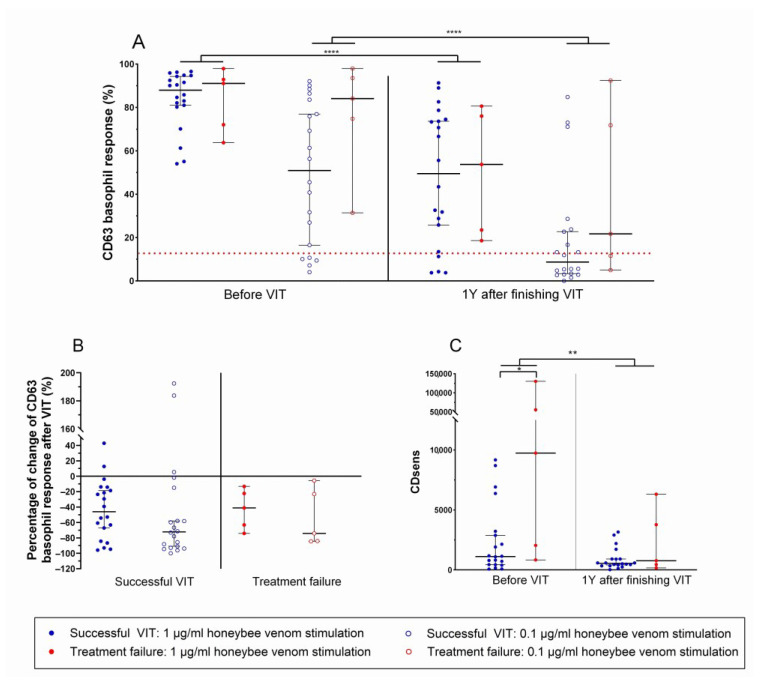
Timeline of immune cellular response in two different groups of patients treated with venom immunotherapy: successful (blue) and treatment failure (red). A basophil activation test reference value of 15% is indicated with a dotted red line. (**A**) Basophil activation test; CD63 basophil response to honeybee venom stimulation, pairwise comparison before and after venom immunotherapy in two different honeybee venom concentrations (1 μg/mL and 0.1 μg/mL). (**B**) Percentage of change in CD63 basophil response to two honeybee venom stimulation concentrations after VIT. (**C**) CDsens calculated from honeybee venom stimulation concentrations of 1 μg/mL, 0.1 μg/mL, 0.01 μg/mL, and 0.001 μg/mL; comparison before and after venom immunotherapy. VIT, venom immunotherapy; *, statistical significance *p*-value ≤ 0.05; **, statistical significance *p*-value ≤ 0.01; ****, statistical significance *p*-value ≤ 0.0001, CDsens; CD-sensitivity.

**Table 1 biomolecules-14-01494-t001:** Clinical and laboratory characteristics of patients.

	All	Successful VIT	Treatment Failure	*p*-Value
No. of patients	25	20	5	/
Age [years], median (range)	49 (23–63)	48.5 (23–63)	49 (26–59)	0.7794
Sex, no. of patients (%)				
Female	3 (12)	3 (15)	0	0.3559
Male	22 (88)	17 (85)	5 (100)	
Specific IgE before VIT [kIU/L], median (range)	6.45 (0.45–52.40)	6.54 (0.45–52.40)	3.46 (0.82–11.40)	0.3766
Initial sting reaction before VIT (Ring and Messmer)				
I	3	3	/	
II	12	10	2	
III	10	7	3	0.4823
Years of VIT, median (range)	6 (5–6)	5.5 (5–6)	6 (5–6)	>0.9999
Systemic adverse events during build-up phase of VIT, n (%)	7 (28)	2 (10)	5 (100)	0.0055

ns, not significant; VIT, venom immunotherapy.

## Data Availability

The raw data supporting conclusions of this article will be made available by the authors on request.

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
