# Peer review of "Cellular and Humoral Response After Induction of Protection and After Finishing Hymenoptera Venom Immunotherapy"

_biomolecules, 2024, doi:10.3390/biom14121494_

Round 1

Reviewer 1 Report

Comments and Suggestions for Authors

The study aimed to evaluate if humoral and ellular biomarkers are associated with the success of Hymenoptera Venom Immunotherapy (VIT). Efficacy of VIT was confirmed by a negative controlled honey bee sting challenge even if this procedure is correlated to safety concerns and logistic problems.

The study was well designed and the research has produced interesting results.

Data are highlighted and presented clearly in Figures and tables.

The Authors report that in their experience serological biomarkers do not reflect VIT induced tolerance. 

Point-to-point revision:

1) At the end of the abstract substitute "two different well-characterized groups of patients..." with two different well-characterized groups of patients, one with successfull VIT and the other with treatment falure".

Line 48: substitute with"is stepwisely increased to the dose of 100 mcg  and the maintenance phase, where 100 mcg are administered at prolonged intervals".

Line86: Basophil activation is also an independent risk factor. 

Author Response

Response to Reviewer 1 Comments

Comments and Suggestions for Authors

The study aimed to evaluate if humoral and ellular biomarkers are associated with the success of Hymenoptera Venom Immunotherapy (VIT). Efficacy of VIT was confirmed by a negative controlled honey bee sting challenge even if this procedure is correlated to safety concerns and logistic problems.

The study was well designed and the research has produced interesting results.

Data are highlighted and presented clearly in Figures and tables.

The Authors report that in their experience serological biomarkers do not reflect VIT induced tolerance. 

Point-to-point revision:

Thank you very much for taking the time to review this manuscript. Please find detailed responses below and the corresponding revisions/corrections highlighted in the re-submitted manuscript.

1) At the end of the abstract substitute "two different well-characterized groups of patients..." with two different well-characterized groups of patients, one with successfull VIT and the other with treatment falure".

Thank you for your valuable comment. In the revised manuscript, we restated the sentence which now reads [lines 26-29 in the revised manuscript]:

 »Our study demonstrated similar sensitization profiles, and humoral and basophil immune responses to immunotherapy in two different well-characterized groups of patients, one with successful VIT and the other with treatment failure.«

Line 48: substitute with"is stepwisely increased to the dose of 100 mcg  and the maintenance phase, where 100 mcg are administered at prolonged intervals".

Thank you for your comment and the consequently improved comprehension of our revised manuscript. The restated sentence now reads [lines 47-49 in the revised manuscript]:

“VIT is a two-step procedure consisting of the build-up phase, in which venom concentration is stepwisely increased to the dose of 100 μg, and the maintenance phase, where 100 μg are administered at prolonged intervals.”

Line86: Basophil activation is also an independent risk factor. 

Thank you for your comment. Line 85 in the revised manuscript has been rephrased.

Thank you again very much for taking the time to review this manuscript and for constructive comments that improved our manuscript.

Reviewer 2 Report

Comments and Suggestions for Authors

To the authors:

1.      General comments:

The original manuscript entitled “Cellular and humoral response after induction of protection and after finishing Hymenoptera venom immunotherapy” shows the humoral and cellular biomarkers measured before during and 1 year after Hymenoptera venom immunotherapy (HVI). In addition, a comparison between responders and non-responders to HVI was carried out. The manuscript is very interesting and novel. I just found minor comments.

2.      Specific comments for revision: b) major comments

1.    Lines 46,53, 69, 72, 77,94. Add a reference.

2.    Line 50, add the number of weeks.

3.    Line 99. Please specify which biomarkers.

4.    Table 1. This should be moved to the results sections, and show the p value for those non-significant comparisons.

5.    Table 1. What is the meaning of “Initial sting reaction”? do you mean before HVI?

6.    Line 216. Add the number of patients.

7.    Figure 2. The sections of the figures must be mentioned in order in the text or change the order of the figures. I would suggest changing Figure 2C-D by Figure 2-E-F.

8.    Lines 285-289. I do not see the relevance and the meaning of the AUC from the venom concentration stimulation.

Author Response

Response to Reviewer 2 Comments

Comments and Suggestions for Authors

1.General comments:

The original manuscript entitled “Cellular and humoral response after induction of protection and after finishing Hymenoptera venom immunotherapy” shows the humoral and cellular biomarkers measured before during and 1 year after Hymenoptera venom immunotherapy (HVI). In addition, a comparison between responders and non-responders to HVI was carried out. The manuscript is very interesting and novel. I just found minor comments.

2.Specific comments for revision: b) major comments

Thank you very much for taking the time to review this manuscript. Please find detailed responses below and the corresponding revisions/corrections highlighted in the re-submitted manuscript.

1.Lines 46,53, 69, 72, 77,94. Add a reference.

Thank you for your careful examination of our manuscript. The references were added [please see lines 45, 49, 52, 69, 76, 93 in the revised manuscript] and have considerably improved the readability of our revised manuscript.

2.Line 50, add the number of weeks.

Thank you for your comment. The time to reach maintenance dose in different protocols has been added [please see lines 49-52 in the revised manuscript] and the sentence now reads:

»The time to reach maintenance dose varies from several hours to several weeks, depending on the protocol used, namely conventional (approx. 50 days), rush (five days), ultra-rush (a couple of hours), or cluster protocol (approx. 30 days)[4,5].«

3.Line 99. Please specify which biomarkers.

Thank you for your comment. The paragraph has been restated for better readability and now reads [please see lines 94-103 in the revised manuscript]:

»Having two very well-defined groups of patients treated with the same protocol in the same referral center, which resulted in either successful VIT or treatment failure according to the sting challenge, enabled us to evaluate the exact dynamics of these below-mentioned biomarkers in both groups and investigate their differences. Thus, we analyzed in detail patients' initial recombinant sensitization profiles and immune responses to VIT by characterizing venom-specific IgE, total IgE levels, BST and venom-specific IgG4 follow-up. We also analyzed basophils` response and basophils` sensitivity to allergen stimulation and different clinical parameters, including systemic adverse events (SAEs) during VIT, in detail.«

4.Table 1. This should be moved to the results sections, and show the p value for those non-significant comparisons.

Thank you for your careful examination of our manuscript. The Table 1 was moved to the results section, and p-values of non-significant comparisons were added as proposed by your valuable comment.

5.Table 1. What is the meaning of “Initial sting reaction”? do you mean before HVI?

The term »initial sting reaction« is used to describe the first allergic reaction which is also indicative for venom immunotherapy. For a better clarification the term was changed to »initial sting reaction before VIT« [please see Table 1 in revised manuscript]. Thank you on your remark.

6.Line 216. Add the number of patients.

Thank you for your valuable comment. The numbers of patients in the reported analysis were added. The sentence now reads [please see lines 218-221 in the revised manuscript]:

»Longitudinal evaluation of patients` blood samples showed a transient increase after reaching maintenance dose (25 patients before vs 23 patients after reaching maintenance dose; p-value = 0.0024) and further a significant decrease (25 patients before vs 25 patients after VIT; p-value < 0.0001) in sIgE to honeybee venom levels throughout VIT (Figure 2A).«

7.Figure 2. The sections of the figures must be mentioned in order in the text or change the order of the figures. I would suggest changing Figure 2C-D by Figure 2-E-F.

Thank you for your valuable suggestion. The Figure 2 has been revised and paragraph titled 3.3. Dynamics of humoral responses are not reflective of tolerance induction changed accordingly.

8.Lines 285-289. I do not see the relevance and the meaning of the AUC from the venom concentration stimulation.

Thank you for your comment. The area under curve analysis made at different venom stimulation concentrations is used as a supporting information to the CDsens analysis. The Figure 3 was revised. The Figure 3C: AUC calculated from honeybee venom stimulation concentration of 1 μg/mL, 0.1 μg/mL, 0.01 μg/mL, and 0.001 μg/mL has been moved to the Supplementary Material Figure S3A.

Thank you again very much for taking the time to review this manuscript and for constructive comments that improved our manuscript.

Reviewer 3 Report

Comments and Suggestions for Authors

Luzar et al. investigated the cellular and humoral response following hymenoptera venom immunotherapy in 25 patients with hymenoptera venom allergy. The researchers evaluated the venom-specific IgE and IgG4, total IgE, and basal tryptase in sera, as well as the basophil activation response to honeybee venom in two groups of patients: those with successful VIT and those experiencing treatment failure. They observed comparable sensitization profiles, as well as humoral and basophil immune responses to immunotherapy in the two distinct groups. The immune responses were characterized by a decrease in sIgE and tIgE levels, an increase in sIgG4 levels, and a reduction in basophil sensitivity, which are characteristic of VIT regardless of its efficacy.  The authors reported that only initial high basophil CDsens and adverse events during VIT demonstrated a significant correlation with VIT outcome and appear to be highly indicative for patients experiencing treatment failure. This study provides some evidence about the cellular and humoral response in VIT. However, I have specific comments that I would like to see addressed.

a) Major comments:

            1. Figure 3A depicts the CD63 basophil response. It would be more informative to present the percentage of CD63 inhibition from before VIT to one year following the completion of venom immunotherapy, and compare the % of CD63 inhibition in successful versus unsuccessful VIT.

2. Figure 3D illustrates the AUC basophil response. As in (1), given that patients exhibit varying responses prior to therapy, it would be more appropriate to present the percentage of AUC inhibition from pre-VIT to one year following the completion of venom immunotherapy, and compare the percentage of AUC inhibition in successful versus unsuccessful VIT.

b) Minor comments:

1.      Throughout the study, in Figures 2 and 3, the statistical comparison bars are not clearly presented, making it difficult to discern the specific comparisons being made (e.g. i) successful versus unsuccessful VIT, ii) successful together with unsuccessful VIT before versus after VIT, iii) successful before versus after, iiii) unsuccessful before versus after). It is recommended to specify the comparisons being made, and each group should be compared individually.

2.     In general, the graph fonts are small; it is recommended to utilize larger font sizes for enhanced clarity.

Author Response

Response to Reviewer 3 Comments

Comments and Suggestions for Authors

Luzar et al. investigated the cellular and humoral response following hymenoptera venom immunotherapy in 25 patients with hymenoptera venom allergy. The researchers evaluated the venom-specific IgE and IgG4, total IgE, and basal tryptase in sera, as well as the basophil activation response to honeybee venom in two groups of patients: those with successful VIT and those experiencing treatment failure. They observed comparable sensitization profiles, as well as humoral and basophil immune responses to immunotherapy in the two distinct groups. The immune responses were characterized by a decrease in sIgE and tIgE levels, an increase in sIgG4 levels, and a reduction in basophil sensitivity, which are characteristic of VIT regardless of its efficacy.  The authors reported that only initial high basophil CDsens and adverse events during VIT demonstrated a significant correlation with VIT outcome and appear to be highly indicative for patients experiencing treatment failure. This study provides some evidence about the cellular and humoral response in VIT. However, I have specific comments that I would like to see addressed.

Thank you very much for taking the time to review this manuscript. Please find detailed responses below and the corresponding revisions/corrections highlighted in the re-submitted manuscript.

  1. a) Major comments:
  2. Figure 3A depicts the CD63 basophil response. It would be more informative to present the percentage of CD63 inhibition from before VIT to one year following the completion of venom immunotherapy, and compare the % of CD63 inhibition in successful versus unsuccessful VIT.

Thank you for your valuable suggestion. The percentage of change of CD63 basophil response to two different honeybee venom concentrations after VIT have been added. However, the statistical comparison between patients with successful VIT and treatment failure did not reach significance. Please see revised Figure 3B.

  1. Figure 3D illustrates the AUC basophil response. As in (1), given that patients exhibit varying responses prior to therapy, it would be more appropriate to present the percentage of AUC inhibition from pre-VIT to one year following the completion of venom immunotherapy, and compare the percentage of AUC inhibition in successful versus unsuccessful VIT.

Thank you for your comment. As suggested by respected opinon of Reviewer 2, the graph presenting AUC calculated from concentrations of 1 μg/mL, 0.1 μg/mL, 0.01 μg/mL, and 0.001 μg/mL; comparison before and after venom immunotherapy, has been moved to the Supplementary Material. Please see Supplementary Figure S3A. Additionally, the percentage of change of AUC after VIT has been added to the Supplementary Material. Please see Supplementary Figure S3B. However, the statistical comparison between patients with successful VIT and treatment failure did not reach significance.

  1. b) Minor comments:
  2. Throughout the study, in Figures 2 and 3, the statistical comparison bars are not clearly presented, making it difficult to discern the specific comparisons being made (e.g. i) successful versus unsuccessful VIT, ii) successful together with unsuccessful VIT before versus after VIT, iii) successful before versus after, iiii) unsuccessful before versus after). It is recommended to specify the comparisons being made, and each group should be compared individually.

Thank you for your comment. We would like to emphasize that only statistical significant comparisons are shown as comparison bars. In Figure 2 no statistically significant differences were observed between groups of successful VIT and VIT failure within the same sampling point (e.g. before VIT, after build-up, or after one year of finishing VIT) [please see lines 324-326, lines 399-401, and lines 408-410 in the discussion section]. Therefore, only comparisons between different sampling times are shown.

Likewise, in Figure 3A, the statistical difference was observed only between different time points and not also within the time point. The only biomarker differentiating patients with successful VIT and treatment failure before VIT was the CDsens showing that patients with treatment failure have markedly higher basophils` sensitivity before the beggining of VIT.

  1. In general, the graph fonts are small; it is recommended to utilize larger font sizes for enhanced clarity.

Thank you for your valuable comment. The graph fonts have been increased.

Thank you again very much for taking the time to review this manuscript and for constructive comments that improved our manuscript.

Round 2

Reviewer 3 Report

Comments and Suggestions for Authors

The authors have satisfactorily addressed all my concerns